# Prognosis in Myelodysplastic Syndromes: The Clinical Challenge of Genomic Integration

**DOI:** 10.3390/jcm10102052

**Published:** 2021-05-11

**Authors:** Tzu-Hua Chen-Liang

**Affiliations:** Hematology and Oncology Unit, University Hospital Morales Meseguer, Marques de los Velez s/n, 30008 Murcia, Spain; tzuchen82@gmail.com

**Keywords:** prognostic impact, clinical–genetic alterations, myelodysplastic syndrome

## Abstract

Myelodysplastic syndromes (MDS) are a group of clonal hematopoietic neoplasms characterized by ineffective hematopoiesis and myelodysplasia with a variable spectrum of clinical–biological features that can be used to build a prognostic estimation. This review summarizes the current most widely used prognostic scoring systems and gives a general view of the prognostic impact of somatic mutations in MDS patients.

## 1. Introduction

Since the publication of the International Prognosis Scoring System (IPSS) in 1997 [1], the international community has focused on finding the best scheme to delineate patient prognosis and cover all subcategories of this group of neoplasms. Nevertheless, with the advent of new massive sequencing tools and their constant improvement, we have a collection of genetic information in terms of the presence of mutations that may change former prognostic scores. Prognostic tools, although designed for most medical conditions, perform better in entities with a heterogeneous clinical course. Myelodysplastic syndromes (MDS) are arguably the most heterogeneous entity among hematological neoplasms. Every day in hematologist waiting rooms, we find patients with the same diagnostic label but very different plans of treatment, from wait-and-watch visits every six months with just an automatized blood count to patients going through the extremely complex road of an allogenic hematopoietic transplantation.

This review summarizes the current most widely used prognostic scoring systems and illustrates the importance of uniting information about different genetic aberrations with clinical features to provide the best understanding of MDS prognostication.

### 1.1. Past and Present MDS Prognostic Models

In daily clinical practice, current prognostic scoring systems are built up from two types of factors: patient (or host) and MDS-related [2]. Patient-related factors may include inherent demographic characteristics as age, gender, ethnicity. Other variables, such as performance status (PS), comorbidities or immune status, are also included here. However, only age and PS are considered to date as correction factors for survival estimation in the most widely used score, IPSS-R. With regard to MDS-related factors, peripheral blood and bone marrow features constitute the main source of predictors. In this category, cytogenetics stands as the most relevant factor in IPSS-R (Figure 1).

Each prognostic score developed in the MDS setting has its strengths and weaknesses. We elaborating about it below, focusing on three still unresolved entities: therapy-related MDS, hypoplastic, and MDS with myelofibrosis [3].

### 1.2. The Classical System and Its Modifications

The International Prognostic Scoring System (IPSS) was established from a combination of previous data on seven scoring systems involving 816 primary MDS patients [1]. Patients who had previously received intensive chemotherapy and those with secondary MDS were excluded. In the final scoring, bone marrow blast percentage, cytogenetic aberrations, and the number of cytopenias were considered. With these variables, authors categorized 4 subgroups: low, intermediate-1, intermediate-2, and high in consideration of the time to acute myeloid leukemia (AML) and overall survival (OS) (Table 1).

The indisputable prognostic value and simplicity of its use made the IPSS to be widely adopted. As it was the cornerstone of MDS outcome prediction for a decade and a half, certain pitfalls were indicated by different groups and researchers. In the pivotal IPSS study, patients were diagnosed according to French–American–British (FAB) criteria. Hence, the final analysis included MDS with 21% to 30% bone-marrow myeloblasts and non-proliferative chronic myelomonocytic leukemia (less than 12,000/µL white-blood-cell (WBC) count), both entities no longer considered among MDS categories in later WHO classifications. Age (60 years or more) did not make it to the final score, as its prognostic significance was demonstrated for OS, but not for time to AML [1]. Other exposed shortcomings included not considering cytopenia depth and not assessing the score value when calculated throughout the course of the disease.

Several modified IPSS models were proposed to improve its applicability, and extend its utility to other groups of patients and disease contexts [3,4]. The WHO scoring system took advantage of the prognostic impact of the diagnostic categories and incorporated the severity of anemia. This score showed us that the number of lineages mattered in the prognostic setting and it was validated not only at diagnosis but also throughout the course of the disease, making it a dynamic score (Table 2).

The MD Anderson’s cancer group designed a scoring system to refine the prognosis of patients included in the low and intermediate 1 IPSS categories, making severe thrombocytopenia a key prediction factor. In addition, they created a comprehensive score system where they incorporated subsets not included (proliferative chronic myeloid monocytic leukemia, therapy-related MDS and previously treated cases) and factors not considered [Eastern Cooperative Oncology Group (ECOG), age, prior red-blood-cell transfusion] in the original IPSS (Table 3) [3].

### 1.3. Revised System

The revised IPSS (IPSS-R) was generated from the evaluation of 7012 patients, and the three classical IPSS building blocks remained: bone-marrow blast percentage, cytogenetics, and cytopenias. Nevertheless, its accuracy was improved both because of a larger population source and because it refined the three blocks. First, the depth of cytopenias was taken into account, adding more points in the final score when anemia, neutropenia, and thrombocytopenia are more severe. Second, cytogenetic categories were further stratified in five subgroups, with the highest stratum showing more weight than that in the bone-marrow blast proportion. Third, the lower blast-percentage categories were divided into two. IPSS-R classifies 5 well separated prognostic categories (very low, low, intermediate, high, and very high) according to OS and time to AML risk (Table 4) [6]. Much like its predecessor, the IPSS-R has become the mainstream score in the MDS field. It is slightly more difficult to calculate than the classical method is, and some online calculators are availableforclinicians (https://www.mds-foundation.org/ipss-r-calculator/, https://www.mdcalc.com/revised-international-prognostic-scoring-system-ipss-r-myelodysplastic-syndrome-mds (accessed on 11 May 2021)).

It soon became apparent that the revised version significantly improved risk stratification in MDS patients [7]. However, in daily clinical routine, therapeutic algorithms in MDS separate patients into two groups, low- and high-risk. With the advent of the revised version and its five categories, it was clear that a unique intermediate category was needed. Pfeilstocker et al. proposed to split the score by a ≤3.5 point cut-off for therapeutical purposes [8].

Some factors were proposed to add independent prognostic value to the IPSS-R: ferritin, hypoalbuminemia, flow-cytometry profile, b2-microglobulin, LDH, or performance status were shown by different groups to be of value [9,10]. Some of these factors showed a low independent predictive weight for OS, but not for time to AML in the IPSS-R pivotal study [6]. Other groups focused their interest on the intermediate category, aiming at its division, allowing for global score stratification in two groups due to the aforementioned routine therapeutic consideration. The use of bone-marrow CD34 positive percentage or the enumeration of blasts exclusively from the myeloid compartment has proven to be useful in some studies [11,12]. Bone-marrow fibrosis (BMF) was an independent factor when confronted with the classical score [13]. In this study of 301 patients with MDS, patients with grades ≥2 BMF had shorter OS and leukemia-free survival (LFS) compared to those with grades ≤1 BMF. The prognostic impact was independent of presence of excess of blasts. Later, BMF was shown to correlate with worse survival within MDS patients who underwent an allogeneic hematopoietic stem cell transplantation [14]. In the IPSS-R era, Ramos et al. reported that advanced myelofibrosis was associated with the presence of mutations in cohesin complex genes and inferior survival [15]. The controversy about when to perform a biopsy (some diagnostic schools avoid the systematic use of this invasive procedure) has precluded this parameter to be assessed in the pivotal series.

## 2. Incoming MDS Prognostic Models

Next-generation sequencing has allowed for large-scale analysis of the molecular profile in MDS. Not surprisingly, its mutational landscape is quite heterogeneous, something that is considered critical to define MDS clinical and pathological features since its first description. The collection of NGS data has granted significant knowledge about the pathophysiological complexity of this blood cancer, involving a set of recurrently mutated genes that play diverse roles in cellular processes, such as DNA methylation, chromatin modification, signal transduction, transcriptional regulation, and RNA splicing [5,7,8,16,17]. Among patients diagnosed with MDS, more than 50% have normal cytogenetics, the only genomic variable included in current prognostic scores, but about 90% harbor mutations. The mutational burden is directly correlated with the number of karyotypic aberrations. Still, a high percentage of low-risk and normal karyotype cases present with an average of 1–3 somatic mutations [18,19].

From a strict quantitative view, the more acquired variants, the more likely the MDS to behave aggressively. Both Bejar and Papaemmanuil showed in a large series of MDS cases that stratification by the number of tumor mutations was highly predictive of LFS. Harboring three or more acquired mutations defined a subset of cases with approximately two years to leukemia transformation [20,21]. Haferlach et al. showed differences in the number of mutations across MDS subtype, from 60% of cases with four or more mutations in the MDS with excess of blasts-2 subgroup to approximately 15% in the MDS with ring sideroblasts subgroup [5].

Another quantitative parameter, the variant allele frequency (VAF), has been studied for prognostic purposes. VAF depicts the percentage of mutated cells within the sample, and it is used as a surrogate marker to define clonal (predominant) or subclonal tumor population. However, for the VAF to be considered as a prognostic factor, the gene affected must be taken into account. *TP53*, *TET2* and *SF3B1* prognostic impact has been shown to be modified by the VAF. On the other hand, there are other targets, i.e., *NRAS* and *EZH2*, that seems to predict for more aggressive course regardless of the VAF [22,23,24,25].

Seven genes were previously reported to harbor independent prognostic significance in MDS and impact on OS: *TP53*, *EZH2*, *RUNX1*, *NRAS*, *ASXL1* and *SF3B1*. All of them, regardless of *SF3B1*, have negative influence on survival and could actively participate as triggers of disease progression to AML [20,26,27]. However, not a single or group of gene mutations were included in the IPSS-R (Figure 1).

*TP53* is a tumor-suppressor gene that plays an essential role in cell-cycle arrest, DNA repair mechanisms, apoptosis induction, and cellular differentiation in response to genetic damage [28]. The mutation of *TP53* is observed in approximately 10–15% of patients with MDS, and predicts for a dismal clinical outcome and poorer responses to treatment [19,29,30]. It is frequently present in complex karyotype cases, where survival is the shortest (less than six months) among MDS patients [28,31,32]. The isolated deletion of the long arm of chromosome 5 is generally considered a favorable feature and key for a good response to lenalidomide. However, it appears that the coexistence of this chromosomal aberration with a TP53 mutation worsens prognosis and contributes to an earlier relapse after treatment with lenalidomide [33,34,35].

Regarding the allele state of *TP53*, a recent study reported robust association between *TP53* biallelic/multihit mutations and an inferior clinical outcome in MDS, in contrast to patients with a unique hit. Strikingly, the authors did not find any difference in OS between patients who had a single hit and those with intact TP53 [25]. Moreover, Sallman et al. found that MDS patients with the *TP53* somatic mutation and VAF of >40% had inferior OS (median 124 days) in comparison with those with VAF < 20% (median OS not reached) [22].

*ASXL1* is mutated in 13–21% of MDS cases, and its presence was described as a predictor of inferior survival in patients with low or intermediate-1 risk according to the IPSS [20]. *SF3B1* is a RNA splicing machinery member, the only gene of which the acquired lesion defines a WHO category (MDS with ring sideroblasts) [36], and its mutational status is associated with improved OS [37], independent after adjustment by IPSS-R group [21]. *RUNX1* is a crucial component of normal hematopoiesis, participating in hematopoietic stem-cell genesis and differentiation. It is mutated in nearly 10% of MDS cases. Patients with a lesion in *RUNX1* have significantly inferior OS compared to those without it [16,20,21,38]. Before NGS was available, it had already been proven that the presence of a *RUNX1* mutation in therapy-related MDS was related to shorter time to AML, but not to impact on OS [39].

As the frequency of cases mutated for a specific gene decreases, associations with a predictive value become blurrier. That is the case, for example, for *IDH1*, *IDH2*, *EZH2*, *CBL* and *U2AF1*, with contradictory results. A peculiar case involves *TET2*, one of the more frequently mutated genes, directly responsible for the impairment of a critical pathway in this disease, but without definitive data regarding its prognostic value and its competence to predict for responses to hypomethylating agents (Table 5).

Using more refined statistical approaches, a recent study addressed the likely underestimated importance of mutation co-occurrence patterns in MDS prognostic settings [63]. Though external validation is needed, the authors stated that specific comutation patterns account for clinical heterogeneity within *SF3B1*, *SRSF2*, and del5q MDS. In addition, acquired mutations may be useful for both anticipating the natural history of the disease at diagnosis, and minimal residual disease assessment and prediction of progression during the course of MDS [64,65,66]. Lastly, the success or failure of emerging targeted therapies (i.e., *IDH1*/*2* or spliceosome inhibitors) may accelerate the implementation of new genes in the workup on MDS patients. Huge effort is currently expended to reach a molecular IPSS-R.

### New Approaches: Machine Learning, Big Data, and “Omics” Integration

Machine learning is a branch of computer science that generates predictive or descriptive models by automatically learning through experience rather than being programmed a priori. Machine learning’s competence to learn from data makes it especially suitable to model complex or nonlinear data. Progress in this field is behind the spectacular advances of recent speech translators and face-recognition applications. Among hematological malignancies, MDS could entail the most complex and heterogeneous clinical and laboratory data. Using machine learning, Nazha and colleagues could improve the discrimination ability of current prognostic scores in the uncharted context of a post-treatment setting [67]. Whether automated algorithms, able to objectively operate with thousands of variables for a single individual, will substitute current scores mostly depends on the accuracy, size, and strict follow-up updates of training databases. International platforms were developed with the aim of gathering clinical–genetic information of patients with blood cancer into one single database from individual cases included in clinical trials and registries. The Harmony alliance, a pan-European public–private partnership, aims to reach statistical power to reveal how molecular data and treatments are intertwined.

To holistically understand the complex clinical and biological disease of MDS, a combination of multiomic data that are able to shed light on the relationship between biomolecules, their function, and their coding and modulating components is imperative. As Winter et al. recently remarked, the integration of immunome (inflammatory cytokines, genes linked to the inflammasome, cellular responses and bone marrow microenvironment factors) with clinical variables is of particular interest in MDS patients, as more data add relevance to the role of immune dysregulation in MDS pathophysiology [68].

## 3. Conclusions and Concerns

Some questions remain for the accurate use of acquired mutations in prognostic systems: Does a specific variant feature (localization, activating or hypomorphic, mono- or biallelic) alter the prognostic value within a very gene? Does the hierarchical place of the mutation (clonal or subclonal) change the prognostic value of an altered gene? How is the prognostic fate of the presence of a mutated gene affected by targeted therapy? Can the co-occurrence of a determined set of mutations predict for a specific outcome? How can we smoothly incorporate comorbidities and performance status into the incoming prognostic models? The answers to these questions will guide us to the final goal of personalized prognostication in a disease that urgently needs new and satisfactory treatments.

## Figures and Tables

**Figure 1 jcm-10-02052-f001:**
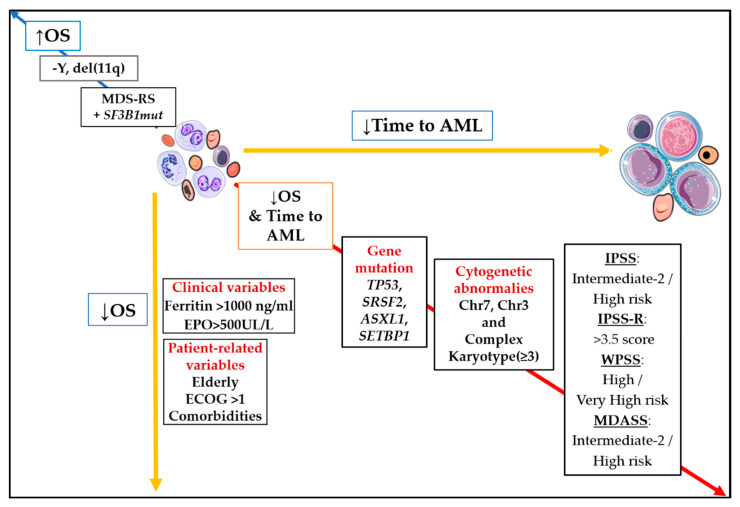
Impact of main prognostic markers on overall survival (OS) and time to acute myeloid leukemia (AML). Diagonal red arrow indicates factors associated both with worse OS and shorter time to AML. Opposite blue arrow is pointing out main factors are associated with better survival. Abbreviations: AML, acute myeloid leukemia; OS, overall survival; Chr, chromosome; EPO, erythropoietin; ECOG, Eastern Cooperative Oncology Group; MDS-RS, myelodysplastic syndrome with ring sideroblasts; −Y, monosomy Y; del(11q), deletion of long arm of chromosome 11; ↓, inferior; ↑; superior.

**Table 1 jcm-10-02052-t001:** International Prognostic Scoring System. Adapted from Greenberg et al., Blood, 1997 [1].

Variable	Parameter	Score	Final Score	Risk Group	LFS Median (Years)	OS Median (Years)
Blasts in bone marrow (%)	<5	0	0	Low	9.4	5.7
	5–10	0.5				
	11–20	1.5				
	21–30	2	0.5–1	Intermediate-1	3.3	3.5
Cytogenetic aberrations	Normal, del(5q), del(20q)	0				
	Other alterations	0.5	1.5–2	Intermediate-2	1.1	1.2
	3 or more alterations, Chrom 7 aberrations	1				
			≥2.5	High	0.2	0.4
Number of cytopenias *	None or 1	0				
	2 or 3	0.5				

* Hemoglobin < 10 g/dL, absolute neutrophile count < 1800/µL, platelets < 100,000/µL. Abbreviations: LFS, leukemia free survival; OS, overall survival.

**Table 2 jcm-10-02052-t002:** World Health Organization classification-based Prognostic Scoring System (WPSS). Adapted from Malcovati et al., Journal of Clinical Oncology, 2007 [5].

Variable	Parameter	Score	Final Score	Risk Group	Cumulative Risk = 0.5 #	
					OS (Month)	Time to AML (Month)
WHO category	RA/RARS/5q–	0	0	Very low	90	NR
	RCMD/RCMD-RS	1				
	RAEB-1	2				
			1	Low	66	NR
	RAEB-2	3				
Cytogenetic aberrations	Normal, del(5q), del(20q)	0				
			2	Intermediate	42	32
	Other alterations	1				
	3 or more alterations, Chrom 7 aberrations	2	3–4	High	30	24
			5–6	Very high	12	6
Transfusion dependency *	No	0				
	Regular	1				

* Red-blood-cell (RBC) transfusion dependency defined as having at least one RBC transfusion every 8 weeks over a period of 4 months. Hemoglobin concentration less than 90 g/L in males and 80 g/L in females was later considered an appropriate surrogate for transfusion dependency. # Approximate values after interpretation of figures from the article of Malcovati et al. In the very low and low risk categories, plateau was reached before 0.5 cumulative risk of AML. Abbreviations: RA, refractory anemia; RARS, refractory anemia with ringed sideroblasts; RCMD, refractory cytopenia with multilineage dysplasia; RCMD-RS, refractory cytopenia with multilineage dysplasia and ringed sideroblasts; RAEB-1, refractory anemia with excess of blasts-1; RAEB-2, refractory anemia with excess of blasts-2; MDS del(5q), myelodysplastic syndrome with isolated del(5q) and marrow blasts less than 5%; NR, nonreached.

**Table 3 jcm-10-02052-t003:** MD Anderson Prognostic Scoring System (MDASS). Adapted from Kantarjian et al., Cancer, 2008 [3].

Variable	Parameter	Score	Final Score	Risk Group	OS Median (Months)
Performance status	>2	2	0–4	Low	54
Age, years	60–64	1			
	≥65	2			
Platelets, ×10^9^/L	50–199	1	5–6	Intermediate 1	25
	30–49	2			
	<30	3			
Hemoglobin, g/dL	<12	2			
Bone marrow blasts, %	5–10	1	7–8	Intermediate 2	14
	11–29	2			
WBC, ×10^9^/L	>20	2			
Karyotype	Chr 7 abnormalities or complex abnormalities (≥3)	3	9–15	High	6
Prior transfusion	Yes	1			

Abbreviations: WBC, white blood cell; OS, overall survival; Chr, chromosome.

**Table 4 jcm-10-02052-t004:** Revised IPSS. Adapted from Greenberg et al., Blood, 2012 [6].

Variable	Score			Final Score	Risk Group	Median Time to AML (Years)	OS, Median (Years)
Blasts in bone marrow (%)	<2		0	≤1.5	Very low	NR	8.8
	>2 to <5		1				
	5–10		2				
	>10		3				
Cytogenetic aberrations	−Y, del(11q)		0				
				2–3	Low	10.8	5.3
	Normal, del(5q), del(12p), del(20q), double including del(5q)		1				
	del(7q), +8, +19, i(17q), any other single or double independent clones		2				
				3.5–4.5	Intermediate	3.2	3
	−7, inv(3)/t(3q)/del(3q), double including −7/del(7q), complex: 3 abnormalities		3				
	Complex: >3 abnormalities		4	5–6	High	1.4	1.6
Cytopenia	Hb (g/dL)	≥10	0				
		8–10	1				
		<8	1.5				
	Platelets (×10^9^/L)	>100	0				
				≥6.5	Very High	0.7	0.8
		50–<100	0.5				
		<50	1				
	ANC (×10^9^/L)	>0.8	0				
		<0.8	0.5				

Abbreviations: AML, acute myeloid leukemia; OS, overall survival; NR, non-reached, Hb, hemoglobin; ANC, absolute neutrophil count.

**Table 5 jcm-10-02052-t005:** Somatic mutations and their association with clinical prognostic impact on outcome of myelodysplastic syndrome.

Pathway	Gene	Specific Group	Clinical Outcome				Reference
			OS	Statistical Approach	Time to AML	Statistical Approach	
Transcription factors	*TP53*		D	Mv	D	Mv	Nazhad et al. [29]; Haase et al. [31]; Bejar et al. [20]
	*RUNX1*		D	Mv	C	-	Bejar et al. [20]
	*BCOR*	All	N	Mv	N	Mv	Damm et al. [40]; Abuhadra et al. [41]
		Frameshift	D	Uv/Mv	D	Uv	
RNA splicing	*SF3B1*	Non-MDS-RS	C	Uv/Mv	C	Uv/Mv	Malcovati et al. [42,43]; Kang et al. [44]
		MDS-RS	I	Mv	C	-	Papaemmanuil et al. [45]
	*SRSF2*		D	Mv	D	Mv	Thol et al. [46]
	*U2AF1*		D	Mv	C	-	Kang et al. [44]
DNA methylation	*TET2*	All	C	Uv/Mv	C	Uv/Mv	Kosmider et.al. [47], Smith et al. [48], Guo et al. [49], Santamaría et al. [50]
		High-risk	N	Mv	D	Mv	Lin et al. [51]
	*IDH1*		C	Uv	C	Uv	Thol et al. [52], Lin et al. [53]
	*IDH2*		D	Uv	C	-	Lin et al. [53]
Chromatin modifiers	*EZH2*		D	Mv	N	Mv	Bejar et al. [20]
	*ASXL1*		D	Mv	D	Mv	Bejar et al. [20], Thol et al. [54]
Cohesin complex	*STAG2*		D	Mv	C	-	Thota et al. [55]
RAS signaling	*NRAS*		C	Uv/Mv	D	Uv	Paquette et al. [56], Murphy et al. [57], Bejar et al. [20]
	*CBL*		N	Uv	C	-	Kao et al. [58]
Others	*SETBP1*		D	Uv/Mv	D	Uv/Mv	Makishima et al. [59], Inoue et al. [60], Fernández-Mercado et al. [61], Damm et al. [62]

Abbreviations: OS, overall survival; AML, acute myeloid leukemia; D, decrease; I, increase; C, conflicting results; N, neutral; Mv, multivariate; Uv, Univariate; MDS-RS, myelodysplastic syndrome with ring sideroblasts. “-“: not available.

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
