# Peer review of "Prognosis in Myelodysplastic Syndromes: The Clinical Challenge of Genomic Integration"

_jcm, 2021, doi:10.3390/jcm10102052_

Round 1
Reviewer 1 Report
This manuscript entitled “The Prognosis in Myelodysplastic Syndrome: The Clinical Challenge of Genomic Integration” summarizes some past and current MDS prognostic scoring systems.
Since many reviews have been published on the MDS prognostic systems in the past, I would suggest that the author focus on the current developments and controversies in MDS prognosis, such as mutation/variant allele frequency, single vs multiple mutations, the degrees of myelofibrosis and dysplasia. The following are the specific comments on the manuscript.
Comment 1: In the abstract, the author should use the WHO definition of MDS, i.e., a group of clonal hematopoietic neoplasms characterized by ineffective hematopoiesis and myelodysplasia.
Comment 2: The figure 1 is confusing and misleading. For example, the cytogenetic abnormalities, such as -7, complex karyotype, have much worse prognosis, and should be placed on far more right than gene mutations on the x-axis.
Comment 3: There are some errors in font, size and words, such as blasts singular vs plural blasts.
Author Response
Reviewing: 1
Comments to the Author
This manuscript entitled “The Prognosis in Myelodysplastic Syndromes: The Clinical Challenge of Genomic Integration” summarizes some past and current MDS prognostic scoring systems.
Since many reviews have been published on the MDS prognostic systems in the past, I would suggest that the author focus on the current developments and controversies in MDS prognosis, such as mutation/variant allele frequency, single vs multiple mutations, the degrees of myelofibrosis and dysplasia.
We acknowledge Reviewer’s major suggestion. We have tried to address those issues as follows:
In section 2, “Incoming MDS Prognostic Models”, adding the following two paragraphs:
“From a strict quantitative view, the more acquired variants, the more likely the MDS to behave aggressively. Both Bejar and Papaemmanuil showed in a large series of MDS cases that stratification by the number of tumor mutations was highly predictive of LFS. Harbouring three or more acquired mutations defined a subset of cases with approximately two years to leukemia transformation[21,22]. Haferlach et al. showed differences in the number of mutations across MDS subtypes, from 60% of cases with four or more mutations in the MDS with excess of blasts-2 subgroup to approximately 15% in the MDS with ring sideroblasts subgroup[16].
Another quantitative parameter, the variant allele frequency (VAF), has been studied for prognostic purposes. VAF depicts the percentage of mutated cells within the sample and it is used as a surrogate marker to define clonal (predominant) or subclonal tumor population. However, for the VAF to be considered as prognostic factor, the gene affected must be taken into account. TP53, TET2 and SF3B1 prognostic impact has been shown to be modified by the VAF. On the other hand, there are other targets, i.e. NRAS and EZH2, that seems to predict for more aggressive course regardless of the VAF [23-26].”
In section 1.2., addressing the number of dysplastic lineages, as follows:
Several modified IPSS models were proposed to improve its applicability, and extend its utility to other groups of patients and disease contexts [3,4]. The WHO scoring system took advantage of the prognostic impact of the diagnostic categories and incorporated the severity of anemia. This score showed us that the number of lineages mattered in the prognostic setting and it was validated not only at diagnosis but also throughout the course of the disease, making it a dynamic score
In section 1.3, addressing concomitant myelofibrosis, as follows:
Bone-marrow fibrosis (BMF) was an independent factor when confronted with the classical score [13]. In this study of 301 patients with MDS, patients with grades ≥2 BMF had shorter OS and leukemia-free survival (LFS) compared to those with grades ≤1 BMF. The prognostic impact was independent of the presence of excess of blasts. Later, BMF was shown to correlate with worse survival within MDS patients who underwent an allogeneic hematopoietic stem cell transplantation [14]. In the IPSS-R era, Ramos et al. reported that advanced myelofibrosis was associated with the presence of mutations in cohesin complex genes and inferior survival [15]. The controversy about when to perform a biopsy (some diagnostic schools avoid the systematic use of this invasive procedure in certain cases) has precluded this parameter to be assessed in the pivotal series.
Comment 1: In the abstract, the author should use the WHO definition of MDS, i.e., a group of clonal hematopoietic neoplasms characterized by ineffective hematopoiesis and myelodysplasia.
Reply 1: We acknowledge your suggestion. We have included it, as follows:
“Abstract: Myelodysplastic syndromes (MDS) are a group of clonal hematopoietic neoplasms characterized by ineffective hematopoiesis and myelodysplasia with a variable spectrum of clinical–biological features that can be used to build a prognostic estimation. This review summarizes current most widely used prognostic scoring systems and gives a general view of the prognostic impact of somatic mutations in MDS patients.”
Comment 2: The figure 1 is confusing and misleading. For example, the cytogenetic abnormalities, such as -7, complex karyotype, have much worse prognosis, and should be placed on far righter than gene mutations on the x-axis.
Reply 2: Reviewer is absolutely right. Though we are not trying to be exhaustive, only orientative, we have used your example to redraw the figure. We think our point is clearer now.
Comment 3: There are some errors in font, size and words, such as blasts singular vs plural blasts.
Reply 3: Corrected.

Reviewer 2 Report
This review is a comprehensive revision of prognostic models and novel biomarkers in MDS.
Comments:
- At the beginning of paragraph 1.1 this sentence is not clear “In daily clinical practice, current prognostic scoring systems are built up from two factor groups: patient- and disease-related”. Please clarify.
- In page 5 the dimension of the font is not equal in all sentences. Please make the text uniform!!
- In figure 1 I do not understand what the red arrow on the left side means. Lower survival in patients with comorbidities, higher ECOG? If so, change position for MDS_RS patients that have a better survival. Also, the other parts of the figure are not complete straightforward. Please arrange it in another way.
- Please expand the section in which you explain potential novel therapeutic approach for each gene mutations. For example, luspatercept in SF3B1 mutated cases and APR-246 in TP53 mutated patients.
Author Response
Reviewing: 2
Comments to the Author
This review is a comprehensive revision of prognostic models and novel biomarkers in MDS.
Comment 1: At the beginning of paragraph 1.1 this sentence is not clear “In daily clinical practice, current prognostic scoring systems are built up from two factor groups: patient-and disease-related”. Please clarify.
Reply 1: We thank Reviewer’s suggestion. We have tried to make this paragraph clearer, as follows:
“In daily clinical practice, current prognostic scoring systems are built up from two type of factors: patient (or host) and MDS-related [2]. Patient-related factors may include inherent demographic characteristics as age, gender, ethnicity. Other variables, such as performance status (PS), comorbidities or immune status are also included here. However, only age and PS are considered to date as correction factors for survival estimation in the most widely used score, IPSS-R. With regard to MDS-related factors, peripheral blood and bone marrow features constitute the main source of predictors. In this category, cytogenetics stands as the most relevant factor in IPSS-R.”
Comment 2: In page 5 the dimension of the font is not equal in all sentences. Please make the text uniform!!
Reply 2:
Corrected
Comment 3: In figure 1 I do not understand what the red arrow on the left side means. Lower survival in patients with comorbidities, higher ECOG? If so, change position for MDS_RS patients that have a better survival. Also, the other parts of the figure are not complete straightforward. Please arrange it in another way.
Reply 3: We acknowledge reviewer to pointing out an issue that also Reviewer-1 stated Though we are not trying to be exhaustive, only orientative, we have redrawn the figure 1 and simplified it. We think our point is clearer now.
Comment 4: Please expand the section in which you explain potential novel therapeutic approach for each gene mutations. For example, luspatercept in SF3B1 mutated cases and APR-246 in TP53 mutated patients.
Reply 4: This is a very important point, which is likely to be comprehensively addressed in a different article of this special issue. To avoid overlap we would wait for the Guess Editor advice and Reviewers consider it should be signed here.
